# Characterisation of Putative Outer Membrane Proteins from *Leptospira borgpetersenii* Serovar Hardjo-Bovis Identifies Novel Adhesins and Diversity in Adhesion across Genomospecies Orthologs

**DOI:** 10.3390/microorganisms12020245

**Published:** 2024-01-24

**Authors:** Intan Noor Aina Kamaruzaman, Gareth James Staton, Stuart Ainsworth, Stuart D. Carter, Nicholas James Evans

**Affiliations:** 1Department of Infection Biology and Microbiomes, Institute of Infection, Veterinary & Ecological Sciences, University of Liverpool, Leahurst Campus, Chester High Road, Neston CH64 7TE, UK; intanaina@umk.edu.my (I.N.A.K.); gstaton@liverpool.ac.uk (G.J.S.); stuart.ainsworth@liverpool.ac.uk (S.A.); scarter@liverpool.ac.uk (S.D.C.); 2Faculty of Veterinary Medicine, Universiti Malaysia Kelantan, Locked Bag 36, Kota Bharu 16100, Malaysia

**Keywords:** bovine leptospirosis, outer membrane protein, adhesins

## Abstract

Leptospirosis is a zoonotic bacterial disease affecting mammalian species worldwide. Cattle are a major susceptible host; infection with pathogenic *Leptospira* spp. represents a public health risk and results in reproductive failure and reduced milk yield, causing economic losses. The characterisation of outer membrane proteins (OMPs) from disease-causing bacteria dissects pathogenesis and underpins vaccine development. As most leptospire pathogenesis research has focused on *Leptospira interrogans*, this study aimed to characterise novel OMPs from another important genomospecies, *Leptospira borgpetersenii*, which has global distribution and is relevant to bovine and human diseases. Several putative *L. borgpetersenii* OMPs were recombinantly expressed, refolded and purified, and evaluated for function and immunogenicity. Two of these unique, putative OMPs (rLBL0972 and rLBL2618) bound to immobilised fibronectin, laminin and fibrinogen, which, together with structural and functional data, supports their classification as leptospiral adhesins. A third putative OMP (rLBL0375), did not exhibit saturable adhesion ability but, together with rLBL0972 and the included control, OmpL1, demonstrated significant cattle milk IgG antibody reactivity from infected cows. To dissect leptospire host–pathogen interactions further, we expressed alleles of OmpL1 and a novel multi-specific adhesin, rLBL2618, from a variety of genomospecies and surveyed their adhesion ability, with both proteins exhibiting divergences in extracellular matrix component binding specificity across synthesised orthologs. We also observed functional redundancy across different *L. borgspetersenii* OMPs which, together with diversity in function across genomospecies orthologs, delineates multiple levels of plasticity in adhesion that is potentially driven by immune selection and host adaptation. These data identify novel leptospiral proteins which should be further evaluated as vaccine and/or diagnostic candidates. Moreover, functional redundancy across leptospire surface proteins together with identified adhesion divergence across genomospecies further dissect the complex host–pathogen interactions of a genus responsible for substantial global disease burden.

## 1. Introduction

Leptospirosis is an emerging, zoonotic bacterial disease affecting a broad spectrum of mammalian hosts worldwide and is caused by spirochetes of the genus *Leptospira*, of which more than 250 pathogenic serovars have been described to date [1]. Leptospirosis has a global distribution and is especially prominent in tropical regions, with human outbreaks resulting in more than one million cases and sixty thousand deaths annually [2]. Further complicating leptospire’s pathogenesis, the *Leptospira* genus is genetically diverse, consisting of a number of genomospecies with variation in their ability to infect different host species [3,4]. Cattle are also susceptible to infection by pathogenic *Leptospira* serovars, and bovine leptospirosis (BL) is a leading cause of reproductive loss and milk drop syndrome worldwide [5,6]. Furthermore, because cattle affected by BL may serve as infection reservoirs, they represent a threat to public health, with those working in farming industries particularly at risk [7,8].

Vaccination is considered a key measure towards limiting the spread of BL and reducing leptospiral burden in healthy cattle. However, most commercially available vaccines approved for use in cattle have been based on bacterin or lipopolysaccharide (LPS) formulations, both of which have been shown to provide limited, short-term protective efficacy against clinical disease via humoral immunity and typically provide a poor spectrum of efficacy against a range of serovars [9].

Outer membrane proteins (OMPs) have been studied as potential leptospiral vaccine candidates due to their expression during natural infection and structural conservation amongst pathogenic *Leptospira* species. The transmembrane OMPs are integral surface proteins which play significant roles in structural integrity and vital physiological functions, including nutrient transport into the cells, adherence to host molecules during bacterial colonisation and the removal of exogenous products [10]. A number of transmembrane *Leptospira* OMPs have been identified and functionally characterised, including the transmembrane porin OmpL1 [11,12,13]. Thus far, such OMP studies have principally focused on *Leptospira interrogans*, an important pathogenic species responsible for leptospirosis in a variety of hosts, including humans, canids and cattle [3]. Here, given most leptospire pathogenesis research has focused on *L. interrogans*, this study aimed to characterise novel OMPs from another important genomospecies, *Leptospira borgpetersenii*, which has global distribution and causes bovine and human disease [14].

The availability of complete genome sequences for a range of pathogenic leptospires allows for investigation into transmembrane OMP-encoding genes through the application of methodologies based on ‘Reverse Vaccinology’ (RV) [15], where key proteins are identified directly from genomes through bioinformatics and subsequently characterised immunologically. From a host–pathogen interaction perspective, such RV approaches also enable functional characterisation, which can also aid vaccine design. In this study, we employed such a strategy to identify unique *L. borgpetersenii* transmembrane OMPs. Once generated, these recombinant putative leptospiral OMPs were functionally and immunologically characterised in vitro. In addition, to dissect the complex host–pathogen interactions of leptospires, we also investigated the adhesion diversity of OmpL1 and one of the novel identified putative OMPs across a variety of genomospecies.

## 2. Materials and Methods

### 2.1. Bacterial Cultures and DNA Extraction

*L. borgpetersenii* serovar Hardjo-bovis strain L550 [15] and *L. interrogans* serovar Copenhageni L1-130 [16] were obtained from the Leptospirosis Reference Centre, Academic Medical Centre, Amsterdam, The Netherlands. Both strains were maintained weekly by passage into sterile Difco™ *Leptospira* Medium Base EMJH (Becton-Dickinson, Detroit, MI, USA) liquid medium, supplemented with 10% (*v*/*v*) Difco™ *Leptospira* Enrichment EMJH (Becton-Dickinson) and grown at 30 °C under aerobic conditions. Late exponential-phase cultures of the spirochetes were subjected to genomic DNA (gDNA) extraction as previously described [17].

### 2.2. In Silico Identification of Unique L. borgpetersenii Hardjo-Bovis OMPs

Several bioinformatics algorithms were applied to predict putative OMP genes considered relevant to *L. borgpetersenii* serovar Hardjo-bovis and which had not been previously subjected to functional characterisation. Complete annotated genomes of *L. borgpetersenii* serovar Hardjo-bovis JB197 and L550 (chromosomes 1 and 2, accession numbers NC_008510 and NC_008511 and NC_008508 and NC_0085509, respectively) [14] were obtained from the National Centre for Biotechnology Information (NCBI). The entire genome of *L. borgpetersenii* serovar Hardjo-bovis L550 was first opened and translated into amino acid sequences using Artemis v17.0.1 (https://www.sanger.ac.uk/tool/artemis/ (accessed on 19 December 2023)); then, it was screened to predict types I and II signal peptides using SignalP version 3.0 (http://www.cbs.dtu.dk/services/SignalP-3.0/ (accessed on 19 December 2023)) [18] to identify features destined for OM insertion or exogenous secretion. Putative β-barrel topology was predicted using three β-barrel prediction programs, PRED-TMBB v2 (http://bioinformatics.biol.uoa.gr/PRED-TMBB/ (accessed on 19 December 2023)) [19], BOMP v1 (http://services.cbu.uib.no/tools/bomp (accessed on 19 December 2023)) [20] and MCMBB v1 (http://athina.biol.uoa.gr/bioinformatics/mcmbb/ (accessed on 19 December 2023)) [21], using default settings. Finally, using a Markov cluster algorithm [22] and BLAST v2.3.0 [23], putative *L. borgepetersenii* OMPs that were shown to share >50% sequence identity with proteins from saprophyte *Leptospira biflexa* serovar Patoc strain Patoc 1 (chromosomes 1 and 2, accession numbers CP_000777 and CP_000778) were excluded from further analysis. Final gene selection criteria were based on (1) the prediction of β-barrel tertiary structure by at least one of the three β-barrel predictor programs, (2) a predicted molecular weight of between 20 and 65 kDa and (3) an OMP domain present in *L. borgpetersenii* serovar Hardjo-bovis but absent from *L. biflexa* serovar Patoc and which had not yet been functionally characterised. Finally, selected genes were submitted to SPAAN v1 [24] to predict their adhesin function.

### 2.3. Cloning, Expression and Purification of Recombinant OMPs

Selected *L. borgpetersenii* serovar Hardj-bovis L550 genes encoding putative transmembrane OMPs and an OMP-positive control (OmpL1 from *L. interrogans* L1-130 [25]) were amplified by High-fidelity DNA polymerase PCR (Thermo Fisher Scientific, Waltham, MA, USA) from the respective strain gDNA extractions using the primers listed (Table 1). Cloning and expression of recombinant proteins were performed using Gateway Technology (Invitrogen™, Carlsbad, CA, USA). PCR products were cloned into a pENTR TOPO vector for directional cloning. Successfully cloned genes were then used to transform *Escherichia coli* TOP10 to propagate the plasmids containing target genes. Target genes were subcloned into the expression vector, pDEST17, which codes an N-terminal polyhistidine tag, using the Gateway™ LR Clonase™ II Enzyme mix (Invitrogen™) and the pDEST17–gene construct propagated in *E. coli* TOP10 cells. All pDEST17–gene constructs were sequenced commercially (Source Bioscience, Nottingham, UK) to ensure the absence of mutations. Protein expression, refolding and purification were performed as previously described [17] with minor modifications. Plasmids containing successfully inserted genes were purified and used to transform the *E. coli* expression strain, BL21-AI (Life Technologies, Paisley, UK), in accordance with manufacturer’s instructions. Strain BL21-AI was selected as it has low basal levels of recombinant protein expression, allowing for the expression of toxic proteins and substantial expression of heterologous genes [26]. Expression cultures were pelleted (5000× *g*; 4 °C 30 min) and resuspended in 50 mM Tris HCl, pH 7.9 (20 mL per 10 g cell paste) with lysozyme (Sigma-Aldrich, Dorset, UK) (5 mg/g cell paste), incubated for 30 min on ice and lysed by sonication (Soniprep-150, MSE, London, UK). Insoluble material was pelleted by centrifugation (10,000× *g*, 4 °C, 30 min) and resuspended in 150 mL buffer solution (50 mM Tris HCl pH 7.9) containing 4% (*v*/*v*) Tergitol (Sigma-Aldrich, Dorset, UK) with rapid stirring for 2 h. Insoluble material was pelleted (10,000× *g*, 4 °C, 30 min) and washed twice in 150 mL Tris HCl pH 7.9 to remove detergent. Inclusion bodies (~500 mg) were solubilised in 6M guanidium hydrochloride, 1 mM EDTA and 50 mM Tris HCl pH 7.9. Solubilised protein was refolded using a modified rapid dilution technique [27] into refolding buffer (50 mM Tris pH 7.9, 250 mM NaCl, 5% (*v*/*v*) LDAO) in a dropwise manner, with rapid stirring for 2 h. Refolded protein solution was dialysed against 0.1% (*v*/*v*) LDAO, 25 mM Tris HCL pH 7.9 and 250 mM of NaCl at 4 °C for 24 h and purified by metal affinity chromatography as described [17]. Eluted protein fractions were analysed by sodium dodecyl sulphate-polyacrylamide gel electrophoresis (SDS-PAGE) to verify purity.

### 2.4. Circular Dichroism Spectroscopy

Purified recombinant proteins were subjected to circular dichroism (CD) spectroscopy to determine secondary structure composition. CD spectroscopic analyses were performed at 20 °C using a Jasco-1100 spectropolarimeter (Jasco, Easton, MD, USA) equipped with a Peltier unit for temperature control. Far-UV CD spectra were measured in a 0.1 mm path-length cell at 0.5 nm intervals. Spectra were collected as an average of three readings from 190 to 260 nm and analysed using CAPITO v1 (https://capito.nmr.leibniz-fli.de// (accessed on 19 December 2023)) software [28]. Minima at either 208 or 215 nm were used to determine the presence of α-helical or β-sheet structures, respectively.

### 2.5. Host Ligand Components

For host ligand interaction studies, a panel of molecules were selected based on those identified as relevant for outer membrane proteins from other spirochetal pathogens [17,29]. The following were purchased from Sigma Aldrich: fibronectin (from bovine plasma), collagen I (from bovine skin), heparin sulphate sodium salt (from bovine kidney), laminin (from Engelbreth-Holm-Swarm murine sarcoma), elastin (from bovine neck ligament and from human skin and aorta), fibrinogen (from bovine plasma) and chondroitin sulphate sodium salt (from bovine cartilage). Bovine serum albumin (BSA) was used as a control.

### 2.6. Binding of Recombinant Proteins to Host Components

Recombinant protein attachment to individual host ECM macromolecules and the plasma component, fibrinogen, was evaluated according to a previously published ELISA protocol with slight modifications [30]. This adapted protocol was previously successfully used to identify spirochetal OMP binding to a range of ECM molecules and was able to identify binding to the same host molecules as the control, leptospire protein OmpL1 [17]. Briefly, non-activated, 96-well microtitre ELISA plates (Microplate Immunlon 2HB 96 well) (ThermoFisher, Horsham, UK) were coated with 100 μL of 5 μg/mL (in PBS, pH 7.2) of host component proteins or with BSA as a control. To allow for coating, plates were incubated for 1 h at 37 °C and then overnight at 4 °C. Unbound protein was removed by washing plates three times (5 min apart) with PBS-Tween 20 (0.05% *v*/*v*-PBST_20_). To prevent non-specific binding of recombinant proteins, wells were blocked with 0.1% (*w*/*v*) BSA solution for 30 min at 37 °C. After washing, 100 μL of his-tagged recombinant leptospiral protein (10 μg/mL in PBST_20_) was added to duplicate wells and incubated for 90 min at 37 °C. Plates were washed six times using PBST_20_ and bound proteins were detected by adding 100 μL monoclonal anti-his tag antibody (Sigma Aldrich, Dorset, UK) (at an optimised dilution of 1:2000) in PBST_20_. Plates were incubated for one hour at 37 °C, and after three washes 100 μL of HRP-conjugated rabbit polyvalent anti-mouse immunoglobulin (at an optimised dilution: 1:10,000) (Sigma Aldrich) was added into the wells and again incubated for one hour at 37 °C. After three washes with PBST_20_, 100 μL of 3,3′,5,5′-tetramethylbenzidine dihydrochloride (TMB) liquid substrate (Uptima-Interchim, Montlucon, France) was added and the plate was incubated at RT for 15 min. Reaction termination was induced by the addition of 100 μL stopping solution (0.5 M hydrochloric acid) per well. Colour absorbance was measured in a microtitre plate reader (Multiskan EX-Thermo Scientific, Wilmington, DE, USA) using a standard 450 nm filter. The ELISA OD values from three independent experiments were averaged for statistical analysis.

### 2.7. Binding-Saturation Experiment

Host components bound by the recombinant proteins under investigation were selected for binding saturation studies. Briefly, ELISA plates were coated with 100 μL of 5 μg/mL (in PBS, pH 7.2) of selected host ECM/plasma components across the wells, incubated and the unbound protein was removed as described above. Next, wells were incubated for 30 min at 37 °C with blocking agent (0.1% (*w*/*v*) BSA in PBST_20_), washed, and 100 μL recombinant protein, diluted in PBST_20_, was added to duplicate wells in increasing concentrations ranging from 0 to 5.0 μM. The addition of monoclonal anti-his tag antibody, conjugated antibody, TMB substrate and stopping solution to the wells was as described for the ECM binding screen. Binding curves were constructed from the mean data of three independent experiments.

### 2.8. Assessment of Recombinant OMPs Binding to Fibrinogen Chains by Far-Western Blotting

Recombinant proteins shown to exhibit specific binding to fibrinogen were subjected to far-Western blotting [31] to confirm the protein–protein interactions and identify the fibrinogen chain structures involved. Briefly, 100 μg/mL of fibrinogen in sample buffer containing 100 mM Tris-Cl [pH 6.9], 200 mM dithiothreitol, 4% SDS, 20% glycerol and 0.2% bromophenol blue was heated at 95 °C for 5 min and loaded into a 4–20% polyacrylamide gradient gel (Bio-Rad, Hemel Hempstead, UK) to separate the fibrinogen chains by electrophoresis. The proteins in the gel were then transferred to a nitrocellulose membrane by electroblotting. The membrane was washed with PBST_20_, stained with Ponceau S stain solution for five minutes, cut into individual strips and blocked with 5% (*w*/*v*) dried skimmed milk for 24 h at 4 °C. After washing with PBST_20_, 30 μg/mL of each his-tagged recombinant OMP solution was added to each strip and incubated at room temperature for 90 min. Strips were washed, incubated with monoclonal anti-his tag antibody and incubated at room temperature for 1 h. After washing with PBST_20_, strips were incubated with HRP-conjugated antibody (polyvalent anti-mouse immunoglobulin) at room temperature for one hour, washed, TMB substrate was added and the strips were covered in a dark room for 15 min. Lastly, strips were dried and photographed using a gel imaging system.

### 2.9. Anti-Leptospiral OMP Antibodies in Cattle Milk Samples

This project was approved by the University of Liverpool ethical review board (Application number: VREC578). Detection of specific antibodies against leptospiral putative OMPs was based on the original ELISA serology method of Yan et al., 1999 [32] with modifications. Briefly, 30 bulk cattle milk samples were analysed which comprised 20 *Leptospira* antibody-positive and 10 *Leptospira* antibody-negative samples, as determined using a commercial *Leptospira* antibody test kit (Linnodee *Leptospira* Hardjo ELISA kit^®^; Balleclare, Northern Ireland) with a sensitivity and specificity of 94.1% and 94.8%, respectively [33]. Samples were randomly selected by a national testing centre (each sample representative of a different farm) at two time points six months apart, with ten leptospire antibody-positive and five controls supplied at each time point. ELISA plates were coated with individual putative OMPs by incubation for one hour at 37 °C and overnight at 4 °C. Unbound protein was removed by washing the plates three times with PBST_20_, and plates were blocked with 100 μL of 0.2% (*w*/*v*) of dried skimmed milk powder solution and incubated at 37 °C for one hour. After plate washing, 100 μL of bulk milk sample was added to the wells and further incubated for one hour at 37 °C. Unbound antibodies were removed through washing three times and bound antibodies were detected by adding 100 μL of mouse anti-bovine immunoglobulin IgG1 or IgG2 (Biorad) (1:1000 in PBST_20_). Plates were incubated again at 37 °C for one hour. After washing, HRP-conjugated anti-mouse IgG antibody, TMB substrate and stopping solution were added as previously described and analysed as in the host component binding experiments. Each assay was repeated as two independent experiments and OD means determined for statistical analysis. For each ELISA, the negative cut-off value was calculated as the control animals’ mean OD plus three standard deviations.

### 2.10. Statistical Analysis

For host component binding analysis, one-way analysis of variance (ANOVA) was performed using Dunnett’s test to compare the binding of recombinant proteins to host molecules against binding to BSA control; a *p*-value of <0.05 was considered statistically significant. For protein binding saturation studies, non-linear regression analysis was performed [34] and the dissociation constant (K_d_) was estimated at half-maximal binding. To screen bulk milk samples for the presence of IgG antibodies to the putative OMPs under investigation, *Leptospira*-positive milk ELISA ODs were compared to those of the control milk samples using the Mann–Whitney U-test (two-tailed), where a *p*-value of <0.05 was considered statistically significant. All results were analysed using Graphpad (Graphpad Prism^®^ version 7.02, San Diego, CA, USA).

### 2.11. In Silico Analysis of the Allelic Diversity of LBL2618 and OmpL1

To identify OMP variants across all leptospiral pathogenic genomospecies for OmpL1 and LBL2618, a BLAST protein search (BLASTp) was performed using query gene sequences from *L. borgpetersenii*. The resulting sequences were compiled into a sequence alignment using Bioedit Sequence Alignment Editor (V. 7.2.5) [35]. All identical sequences were removed and locus tags were examined to verify annotation accuracy. Phylogenetic trees of the LBL2618 and OmpL1 variants were constructed using the maximum-likelihood method [36]. The respective trees, together with generated sequence identity matrices, were analysed for each protein so that a total of five sequences for each protein (including the original *L. borgpetersenii* serovar Hardjo-bovis L550 sequence) were selected from different separated deep branches.

### 2.12. Construction of Expression Vector

Signal peptides were removed from the LBL2618 and OmpL1 variant amino acid sequences and submitted to GeneMill (University of Liverpool, Liverpool, UK) for synthesis within pET-21a (+) expression constructs.

### 2.13. Protein Expression and Purification of Leptospira Recombinant OMP Variants

All recombinant protein expression was performed as described above, with minor modifications. In large scale expression, the transformed BL21-AI *E. coli* cultures were induced using a combination of 0.2% (*w*/*v*) L-arabinose and 1 mM IPTG, as recommended by the manufacturer. Inclusion body extraction and protein purification were as described above. All expressed OmpL1 and LBL2618 variants were analysed by SDS-PAGE and CD, to confirm purity and secondary structure composition, respectively, before undergoing host ligand binding analyses.

## 3. Results

### 3.1. Selection of Putative Transmembrane OMPs from L. borgpetersenii Serovar Hardjo-Bovis L550 Genome

The in silico identification of OMPs using various bioinformatics-based RV approaches led to the selection of six previously uncharacterised, hypothetical transmembrane OMP-encoded genes from *L. borgpetersenii* serovar Hardjo-bovis L550 (Table 2). OmpL1 from both *L. interrogans* serovar Copenhageni Fiocruz L1-130 and *L. borgpetersenii* serovar Hardjo-bovis strain L550 were selected as positive controls for immobilised host ligand binding assays and bulk milk tank antibody assays, respectively, since *L. interrogans* OmpL1 host ligand attachment has been previously characterised [11,13] and is recognised by host antibodies within human and canine serum [37,38]. 

### 3.2. Overexpression, Purification and Determination of Secondary Structure of Recombinant OMPs

All genes were successfully cloned; however, only three of the novel genes and the positive controls (LBL_0375, LBL_0972, LBL_2618, LBL 2510 and LIC_10973) were successfully expressed as inclusion bodies and subsequently refolded and purified. The final concentration of the recombinant purified proteins ranged from 0.5 to 2.0 mg/mL; preparations were of sufficient purity (Figure 1) and remained stable in solution at 4 °C.

Structural composition analysis of the purified proteins using CD spectroscopy (Figure 2) showed that the OmpL1 proteins (rLBL2510 and rLIC10973) and the novel leptospiral protein rLBL2618 consisted predominantly of β-sheets with spectral minima occurring at 215 nm and peaking at ~195 nm, which is a distinctive feature of β-barrel OMP structures. One novel protein (rLBL0972) exhibited a CD spectrum with minima at 208 and 215 nm, consistent with the presence of both α-helical and β-sheet structures. Intriguingly, one recombinant protein (rLBL0375) (Figure 2D) exhibited a substantially different spectrum, which was consistent with the presence of α-helical structures only.

### 3.3. Assessment of Protein Structure Using SDS-PAGE Analysis

Upon SDS-PAGE analysis, the OmpL1 preparations were observed to contain two bands (~25 kD and ~31 kDa), suggesting the presence of different fold states. We therefore sought to confirm that the OmpL1s expressed here were heat-modifiable, as reported by Shang et al. (1995) [13]. Both recombinant OmpL1s required heating to 100 °C for 40 min together with 10 mM DTT and 8 M urea for the protein to migrate to its denatured size of 31 kDa rather than the further migrated folded band of 25 kDa (Figure 3). For rLBL0972 and rLBL2618, no heat-induced changes in band size were observed. SDS-PAGE analysis of purified rLBL0375 inclusion bodies consisted predominantly of a 37 kDa protein. However, upon refolding, this protein migrated to a position consistent with an MW of ~55 kDa (Figure 3, Lane 4) and was not heat-modifiable.

### 3.4. Adherence of Recombinant OMPs to Host Components

As transmembrane leptospiral OMPs are surface-exposed and potentially interact with the host, we attempted to evaluate the binding function of *L. borgspetersenii* L550 putative OMPs to a range of host ECM and serum components. As shown in Figure 4B, rLBL2618 showed significant binding to fibrinogen, laminin and fibronectin (*p* > 0.001); rLBL0972 bound to both fibrinogen (*p* < 0.001) and fibronectin (*p* < 0.05) (Figure 4C); and rLBL0375 bound only to fibrinogen (*p* < 0.01) (Figure 4D). An analysis of the OmpL1 of *L. interrogans* L1-130 revealed significant binding to fibrinogen, laminin and fibronectin (*p* < 0.001), as well as chondroitin sulphate (*p* < 0.05). No significant binding interactions were observed between the novel recombinant proteins and chondroitin sulphate, heparan sulphate or collagen.

To characterise these interactions further, binding was measured as a function of recombinant protein concentration (from 0 to 5.0 μM) to calculate the dissociation constant (K_d_) (Figure 5 and Table 3).

For rLBL2618, high-strength binding interactions were achieved for fibrinogen (K_d_ 0.05 ± 0.01 μM), fibronectin (K_d_ 0.10 ± 0.02 μM) and laminin (K_d_ 0.14 ± 0.01 μM) (Figure 5A, Table 3). Additionally, a binding interaction with BSA was observed for rLBL2618, although it did not saturate. For rLBL0972, comparatively strong interactions with fibrinogen (K_d_ 0.20 ± 0.06 μM) and fibronectin (K_d_ 0.44 ± 0.21 μM) were observed (Figure 5B, Table 3). Despite the significant binding to fibrinogen observed for rLBL0375 (Figure 4D), this protein did not demonstrate saturable binding to fibrinogen and this interaction was considered non-specific (Figure 5C). The positive control, the OmpL1 of *L. interrogans*, exhibited saturable binding to fibronectin (K_d_ 0.81 ± 1.30 μM), fibrinogen (K_d_ 0.92 ± 1.38 μM) and laminin (K_d_ 1.46 ± 5.44 μM) (Table 3), whilst binding to chondroitin was considered non-specific as binding saturation was not achieved.

### 3.5. Recombinant Leptospiral OMPs Binding to Fibrinogen Components

To further define the interactions observed between recombinant leptospiral OMPs and fibrinogen, a far-Western blot was used (Figure 6). Differential binding (based on the intensity of the visible band) was observed to fibrinogen with rLBL2618 > rLBL0972 > rLBL0375, where rLBL0375 exhibited a very weak interaction with the β- and ɤ-chains, whilst rLBL2618 and rLBL0972 bound to all three chains (α, β, γ).

### 3.6. Evaluation of Host Antibody Reactivity against Leptospiral Proteins

The immunogenic properties of the leptospiral recombinant proteins were determined by assessing whether they were recognised by IgG antibodies present in bulk milk samples collected from animals known to have been previously exposed to bovine leptospirosis. As depicted in Figure 7, the control protein, OmpL1 (LBL_2510) (Figure 7A) of *L. borgpetersenii* serovar Hardjo-bovis L550, and two of the novel proteins, rLBL0972 (Figure 7C) and rLBL0375 (Figure 7D), were recognised by IgG1 antibodies in the milk of naturally infected cattle.

When a negative/positive cut-off value was applied, anti-rLBL0375 IgG1 antibodies (the most significant association) were detected in 72.2% of the *Leptospira*-positive bulk milk tank samples. The IgG1 antibodies to all leptospiral proteins were used to determine if binding to one correlated with binding to another using Pearson’s coefficient correlation. Antibodies to all newly identified putative OMPs correlated with each other as represented by weak to strong r values (ranging from +0.35 to +0.80, *p* < 0.001, *p* < 0.0001). In contrast, there was no such correlation between antibodies to these putative OMPs and OmpL1 from either genomospecies. Conversely, these data demonstrated that the milk IgG1 antibody titres against the OmpL1s from both *L. interrogans* and *L. borgpetersenii* positively correlated (r = 0.41, *p* < 0.0001) (Table 4).

### 3.7. In Silico Analysis of the Allelic Diversity of LBL2618 and OmpL1

#### Selection of OmpL1 and LBL2618 across Pathogenic Genomospecies

Five amino acid sequences of OmpL1 and five amino acid sequences of rLBL2618, representing the diversity across all pathogenic *Leptospira* genomospecies, were selected from different distinct representative deep branches of the phylogenetic tree (Figure 8A,B). A CLUSTALW alignment [39] showed that the shared amino acid sequence identity for OmpL1 was ≥85%, in line with a previous study [40]. The amino acid sequence identity for LBL2618 was ≥65%, with all sequences annotated with Domain of Unknown Function 1566 (DUF1566).

### 3.8. Variant OmpL1 and rLBL2618 Production, Secondary Structure and Binding to Host Molecules

One putative OMP (*L. santarosai* rLBL2618) failed to express and was excluded from the study. The remaining purified recombinant OmpL1 and rLBL2618 purified proteins, when analysed by CD, were shown to comprise a predominantly β-sheet secondary structure, consistent with a folded state (Appendix A).

All recombinant proteins were subjected to the functional ECM binding screens described above, revealing considerable variation in ligand specificity between alleles. For instance, the OmpL1s of *L. borgpetersenii* serovar Hadjo-bovis L550 and *L. santarosai* both adhered to bovine fibrinogen (*p* < 0.001) and bovine elastin (*p* < 0.001), whilst the OmpL1 of *L. santarosai* additionally bound to laminin (*p* < 0.001) and chondroitin (*p* < 0.05). The OmpL1s from both *L. interrogans* also showed significant host molecule binding variations: *L. interrogans* serovar Copenhageni L1-130 had additional binding preference for fibronectin (*p* < 0.001), laminin (*p* < 0.001) and chondroitin (*p* < 0.05), whereas the OmpL1 of *L. interrogans* serovar Pyrogenes bound only to fibrinogen (*p* < 0.001). No significant binding to collagen and heparan sulphate by any of the OmpL1s was observed (Figure 9).

For the rLBL2618 variants, there was also some variation in host ligand binding between species. In general, all rLBL2618 proteins bound to fibrinogen (with *p* < 0.001 to *p* < 0.05). The rLBL2618 proteins of *L. alstonii* and *L. borgpetersenii* serovar Hardjo-bovis L550 exhibited additional bindings to laminin and fibronectin (*p* > 0.001) (Figure 10). No significant binding interactions were observed to heparan sulphate, chondroitin, collagen or elastin.

### 3.9. Binding Saturation Curves of Leptospiral Recombinant OMP Variants to Select Host Molecules

All OmpL1 and LBL2618 variants for which significant host molecule binding occurred were selected for further analysis to assess whether these interactions were concentration-dependent and saturable using the aforementioned ELISA-based assay. For OmpL1, all proteins showed a saturated binding interaction to fibrinogen (Table 5). Estimated K_D_ values for this interaction were observed to be relatively similar for both *L. santarosai* and *L. alstonii*, with values of 0.28 ± 0.07 μM and 0.21 ± 0.12 μM, respectively. The OmpL1 from *L. borgpetersenii* serovar Hardjo-bovis L550 showed a stronger affinity for bovine elastin (K_D_ 0.82 ± 0.35 μM) compared with *L. santarosai* (K_D_ 1.72 ± 1.00 μM) (Table 5). The binding affinities for the interactions between the OmpL1s of *L. santarosai*, *L. borgpetersenii* and *L. interrogans* serovar Copenhageni L1-130 and fibronectin and laminin were considered to be relatively weak given that binding saturation occurred at >1.0 μM. Binding saturation to chondroitin by both *L. santarosai* and *L. interrogans* serovar Copenhageni L1-130 OmpLs was not achieved, and the binding interactions were thus considered non-specific (Table 5). For all rLBL2618 variants, binding variations were observed between species, with *L. borgpetersenii* serovar Hardjo-bovis L550 and *L. alstonii* sharing a similar binding profile to several host molecules (fibrinogen, laminin and fibronectin, *p* < 0.001). All genomospecies orthologs achieved binding saturation to the host molecules identified as having significant binding in the initial binding screen (Table 5). Interestingly, all rLBL2618 variants demonstrated saturable and specific binding to the control BSA.

## 4. Discussion

The adherence of pathogens to susceptible host cells and tissues is a critical step in the establishment of infection. Like most invasive bacteria, the leptospiral cell surface plays an important role in this infection process. To date, several leptospiral putative surface-exposed OMPs have been discovered and characterised, although these were mostly focused on a single pathogenic species (*L. interrogans*) [10]. Another important pathogenic species, *L. borgpetersenii*, associated with disease in both cattle and humans, appears to have more than 300 unique genes, greater than 50% of which have no assigned function [41]; they are therefore worthy of consideration as both mediators of host–pathogen interactions and as potential vaccine candidates. In a previous study, 238 of the *L. borgpetersenii* proteins predicted as surface-exposed were immunologically evaluated in hamster infection models [42], although none of the recombinant proteins or protein fragments investigated were found to protect against renal colonisation, nor were their roles in host–pathogen interactions scrutinised. Moreover, given the importance of conformational epitopes in eliciting a protective immune response [43], it is possible that the use of unfolded recombinant proteins abolished their protective immunogenic properties. Here, we identified six novel putative OMP-coding genes, three of which were successfully produced as recombinant proteins, refolded and subsequently characterised, together with relevant OmpL1 controls. Several factors might cause variation in recombinant expression success across these different novel proteins, such as toxicity, metabolic stress and pH imbalance resulting from protein overexpression, and these are typically protein-specific [44]. When analysed by CD spectroscopy, two of the novel recombinant *L. borgpetersenii* putative OMPs (rLBL0972 and rLBL2618) and both OmpL1 control proteins yielded spectra consistent with a β-sheet secondary structure, suggesting that protein folding had occurred and therefore supporting their predicted classification as surface-exposed transmembrane proteins, many of which are known to be adhesins and in agreement with the predictions in Table 2. An analysis of the recombinant OmpL1 proteins from both pathogenic species by CD spectroscopy and an electrophoretic mobility assay confirmed that these proteins exhibited characteristics shared with porins, including a tightly folded conformation and heat stability [13]. Interestingly, rLBL0375 exhibited CD spectra indicative of a substantial α-helical composition, which contrasts with the expected β-sheet topology predicted in silico. Interestingly, rLBL0375 exhibited an SDS-PAGE ‘gel-shifting’ characteristic as it did not migrate to its expected molecular weight after refolding. This anomaly is in agreement with the CD spectra, and together, these data suggest that this protein is composed predominantly of an α-helical secondary structure, with detergent/α-helical membrane protein interactions known to cause reduced gel mobility [45]. No changes in band size in terms of heat modification or reduced gel mobility were observed in rLBL0972 and rLBL2618. Nevertheless, all proteins were stable in solution, which indicate these proteins had refolded to their native structures.

Through functional studies, these proteins exhibited binding properties toward multiple host macromolecules (with the exception of rLBL0375), as indicated by both concentration-dependent and saturable binding curves, which indicates they play a role in host adherence, a key pathogenic mechanism. The specific, saturable binding to fibrinogen for the majority of OMPs is unsurprising because several known *L. interrogans* proteins have been demonstrated to bind to fibrinogen in vitro [46]. Fibrinogen is a plasma protein essential for haemostasis and wound repair. The binding of a bacterial cell to fibrinogen may lead to the disruption of blood coagulation, causing haemorrhage, which is a common feature of severe leptospirosis manifestation in humans [47]. Moreover, where saturable binding was identified, far-Western blotting analysis further characterised the binding of the novel OMPs and demonstrated their capacity to adhere to all three fibrinogen chains, akin to the interaction profile for OmpL1 with fibrinogen [17]. Contrastingly, the observed varying degrees of binding affinity to fibrinogen across the OMPs could indicate differences in binding mechanisms, steric hindrance or enable different fibrinogen uses.

In addition to fibrinogen, our data also demonstrated that both rLBL0972 and rLBL2618 bound to fibronectin, whereas rLBL2618 additionally bound to laminin. These findings are comparable to other previously characterised leptospiral proteins, including *ligs* (LigA and LigB), *lens* (B-F) and several Lsa proteins (Lsa66, Lsa21 and Lsa30) [48]. Host adhesion to these molecules is considered crucial to enhance invasion and dissemination to begin bacterial colonisation of the host. Notably, one putative OMP (rLBL2618) demonstrated concentration-dependent binding to the control protein BSA. Interestingly, an interaction with BSA has been reported previously for the leptospiral OMP, OmpL37 [12], suggesting that BSA may be a biologically significant leptospiral target. Further studies are warranted to explore this interaction.

The binding of different OMPs to similar ligand profiles in this study indicates considerable functional redundancy between these cattle *L. borgpetersenii* OMPs. Indeed, having multiple OMPs encode abilities to bind multiple ligands may allow for immune evasion or allow for different bacterial advantages depending on location or disease stage. Functional redundancy has been widely described for many bacterial pathogen surface proteins [49], and the phenomenon is considered typical in *Leptospira* spp. [41]. This redundancy has been attributed to the process of genomic expansion through gene duplication [14]. Surprisingly, our results revealed that even though the *L. borgpetersenii* species underwent genome reduction resulting in restricted host transmission [14], this species maintained OMPs unique to this genomospecies which enhance functional redundancy. Pathogenic leptospires have more paralogs compared with the saprophytic representatives [50], which likely explains the occurrence of functional redundancy within *L. borgpetersenii*. However, the temporal expression of these proteins is unknown, and they may appear and operate differently in certain disease stages or in different tissues [41].

Because a positive correlation exists between the amount of milk IgG1 antibodies generated against all three novel leptospiral proteins under investigation, it is probable that these proteins are expressed by *L. borgpetersenii* during natural infection and are recognised by the host immune system. Conversely, milk IgG2 reactivity against all OMPs was comparatively low; this could be due to the preferential uptake of IgG2 back to the host extracellular fluid, restricting passage into the alveolar lumen and subsequently into the mammary secretions [51]. Interestingly, milk IgG1 antibody titres against rLBL0972, rLBL0375 and an immunological positive control (*L. borgpetersenii* OmpL1) exhibited a statistically significant association with leptospirosis disease status. As previously described, OmpL1 has been demonstrated as a promising diagnostic antigen as it is recognised by the immune system (serum antibodies) of infected human patients during both acute and chronic leptospirosis [11,37], but it has never been used to test for anti-leptospiral antibodies in cattle milk. Notably, in our assay, the correlation between milk IgG1 anti-rLBL0375 antibodies and disease status was statistically superior to the correlation between anti-OmpL1 IgG1 antibodies and disease status. This shows that this protein may be useful as a diagnostic antigen to assign disease status in bovine leptospirosis, particularly when assaying milk samples. Thus, although we were not able to classify its function, this result suggests that rLBL0375 is likely expressed in cattle milk during infection. Similarly, another leptospiral lipoprotein (LipL41) also has no apparent binding function, but is recognised by the host immune system [37,52]. Whilst we were only able to identify direct statistical disease associations for a subset of proteins, given the variability in the data investigated it may be that the remaining proteins could also exhibit statistical significance if a larger sample size is used in the future, although for diagnostic potential the current analysis used has allowed for the discovery of potential diagnostic antigens and enabled the preclusion of other proteins as they are unlikely to enable high specificity and sensitivity. Moreover, positive correlations of IgG1 cattle milk titres were observed between the novel OMPs but not when these OMPs were compared with OmpL1. Such positive correlations between these novel OMPs may indicate that rLBL2618, rLBL0972 and rLBL0375 interact with the host in a similar fashion, such as being expressed during a similar time/immunological period of infection or by sharing physiochemical properties different to OmpL1. A moderate antibody correlation between the two OmpL1s suggests conserved immunogenicity given they share similar OMP domains but are from different pathogenic *Leptospira* species, as previously reported [53,54], especially given there is a >85% conservation between the OmpL1s of *L. interrogans* and *L. borgpetersenii*. Interestingly, *Leptospira* was previously detected in the mammary tissues of rats [55], leading to vertical transmission; therefore, rLBL2618, rLBL0972, rLBL0375 and OmpL1 may be directly involved in *Leptospira* colonisation of host mammary tissues. Moreover, OmpL1 was shown in hamsters to be an effective vaccine candidate against heterologous *Leptospira* challenge [56] and, therefore, further studies are recommended to evaluate the effectiveness of rLBL2618, rLBL0972 and rLBL0375 as vaccine components. Whilst it is still unclear whether IgG1 antibodies against these surface proteins correspond to leptospirosis protection, indications from other studies suggest future protection studies with these proteins are now needed to characterise any protective immune responses, especially as Ompl1 has been shown to have substantial IgG1 responses [11] and also to have protective efficacy when used as a vaccine [53,56].

Since the OmpL1s of *L. borgspetersenii* and *L. interrogans* exhibited marked diversity in their ability to adhere to host molecules, we investigated whether this diversity existed across a range of leptospiral species. Considerable species-specific variation in the binding profiles was found to exist across the OmpL1 alleles, although all alleles bound significantly to fibrinogen. *L. interrogans* serovar Pyrogenes and *L. noguchii* failed to bind to any other host molecule tested, whilst the OmpL1s from *L. borgspetersenii* serovar Hardjo-bovis L550 and *L. santarosai* bound to multiple host ligands, including elastin. To our knowledge, this is the first study to demonstrate OmpL1 interactions with elastin. Given that humans have been demonstrated to be susceptible to infection by both *L. borgspetersenii* and *L. santarosai* [57,58] and that abraded skin is an exploited point of entry to their hosts, these data implicate elastin as a potential target during zoonotic transmission events.

Similar diversity in adhesion ability was observed for rLBL2618. For instance, rLBL2618s from both *L. borgspetersenii* serovar Hardjo-bovis L550 and *L. alstonii* exhibited identical binding profiles, binding to fibronectin, laminin and fibrinogen, whereas the orthologs identified in *L. kirschneri* and *L. noguchii* bound to fibrinogen only.

We consider that such differences in binding function for OMPs in different *Leptospira* species and serovar could be the key to their adaptation or survival in the hosts. For instance, serovar *Copenhageni* can infect a wide range of hosts, including dogs, rats and man [59,60,61], whereas serovar Pyrogenes is commonly found in cattle reservoirs [62]. It should be noted that the distribution of pathogenic *Leptospira* serovars maintained by an animal host is often geographically unique, such as serovar Sokoine in African cattle and serovar Sarawak in Malaysian cattle [63,64]. This may be possible because OmpL1 is an important leptospiral adhesin contributing to host binding preference as they are one of the few proteins that can only be found in pathogenic species [3].

All rLBL2618 proteins are bound to fibrinogen and the majority are bound to laminin and fibronectin, respectively. Interestingly, we found that both the OmpL1 and rLBL2618 of *L. noguchii* only bound to fibrinogen. It is not known whether the limited binding to host molecules may have an effect on the transmission of the bacteria to other hosts (e.g., host restriction), or if it could be that the species is restricted to certain host tissues. Also of note is that despite a higher sequence conservation of 85% for OmpL1 compared with 65% for rLBL2618 across genomospecies, a greater diversity in the number of ligands bound by OmpL1 was observed. Moreover, there was a greater variation in binding affinities to identified binding ligands for OmpL1 which could also impact host/pathogen interactions, changing ligand preferences if all host ligands are available or potentially changing the use of the host molecule. Such differences may be due to variations in the quantity and type of surface structures presented to the host together with differences in immune selection pressure. A greater immune selection may act on OmpL1, resulting in immune interactions, with ligand binding surface loop regions changing substantially and modifying adhesion ability, whilst the OmpL1 β -barrel component sits protected within the outer membrane and is highly conserved. Whist we were unable to express one rLBL2618 variant, likely due to individual ortholog compositions, even if variants are restricted to identical genomospecies for OmpL1 and rLBL2618, there is still a greater variation in ligands bound for OmpL1, outlining the substantial functional plasticity of this important leptospire surface protein.

## 5. Conclusions

Taken together, we consider that the novel proteins identified in the present study may play a significant role during leptospiral invasion of the host and should be considered for future vaccine and diagnostic development. Furthermore, by examining the allelic functional diversity that exists for OmpL1 and LBL2618, we begin to shed light on the molecular interactions underpinning host colonisation events. Further studies are needed to better define the molecular mechanisms that underpin the multifunctionality of leptospiral OMPs.

## Figures and Tables

**Figure 1 microorganisms-12-00245-f001:**
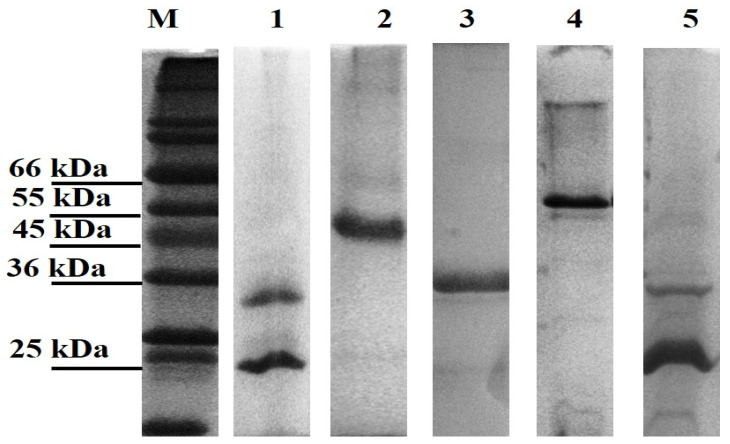
Leptospiral recombinant putative OMP purifications analysed on 12% (*w*/*v*) SDS-PAGE gel under denaturing conditions. Each lane represents a protein encoded by the selected gene; Lane 1: LBL_2510 (OmpL1), Lane 2: LBL_2618, Lane 3: LBL_0972, Lane 4: LBL_0375 and Lane 5: LIC_10973 (OmpL1-positive control). Lane M represents standard protein markers.

**Figure 2 microorganisms-12-00245-f002:**
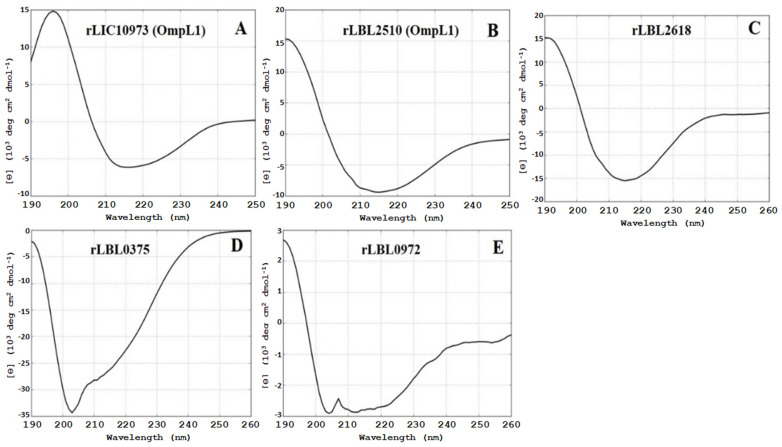
(**A**–**E**): Circular dichroism (CD) spectra of the recombinant OMPs depicting a predominance of β-sheet secondary structure, with the exception of rLBL0375 (**D**), which exhibited a mixed α-helical structure. Far-UV CD are represented as an average of three scans recorded from 190 to 260 nm.

**Figure 3 microorganisms-12-00245-f003:**
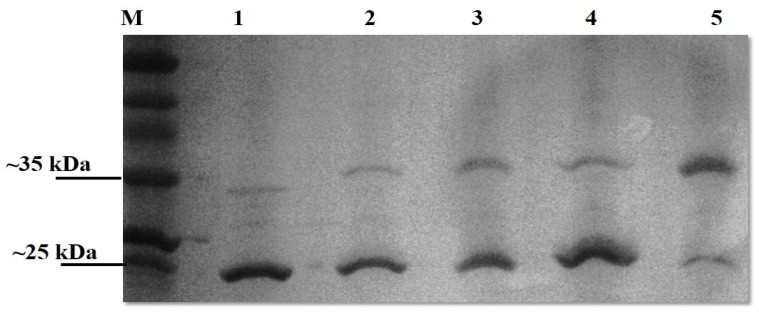
The effect of heat, DTT and urea on a recombinant OmpL1 (rLIC10973). Denatured and native molecular masses are 31 kDa and 25 kDa, respectively. The OmpL1 samples were subjected to five different treatments; Lane 1: unheated samples at 25 °C, Lane 2: heated samples at 100 °C for 10 min with 10 mM DTT, Lane 3: heated samples at 100 °C (10 min) with 10 mM DTT and 8 M urea, Lane 4: heated samples at 100 °C (40 min) with 10 mM DTT, and Lane 5: heated samples at 100 °C (40 min) with 10 mM DTT and 8 M urea. Lane M represents standard protein markers.

**Figure 4 microorganisms-12-00245-f004:**
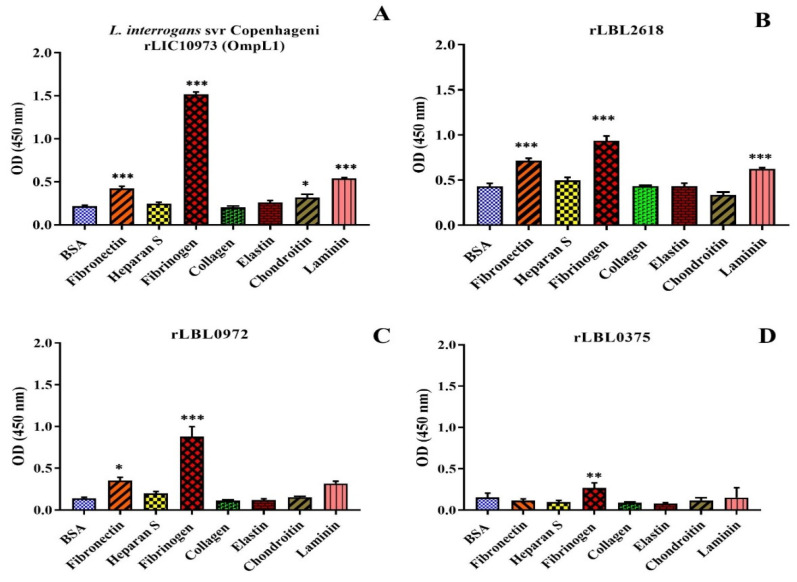
(**A**–**D**): Binding of novel leptospiral recombinant OMPs and control protein (OmpL1 of *L. interrogans* serovar Copenhageni L1-130) to host molecules. Data represent the mean absorbance at 450 nm ± SEM of three independent experiments. The binding of leptospiral recombinant OMPs to host molecules was compared by one-way ANOVA analysis followed by Dunnett’s multiple comparison test using BSA as a control (*** *p* < 0.001, ** *p* < 0.01, * *p* < 0.05).

**Figure 5 microorganisms-12-00245-f005:**
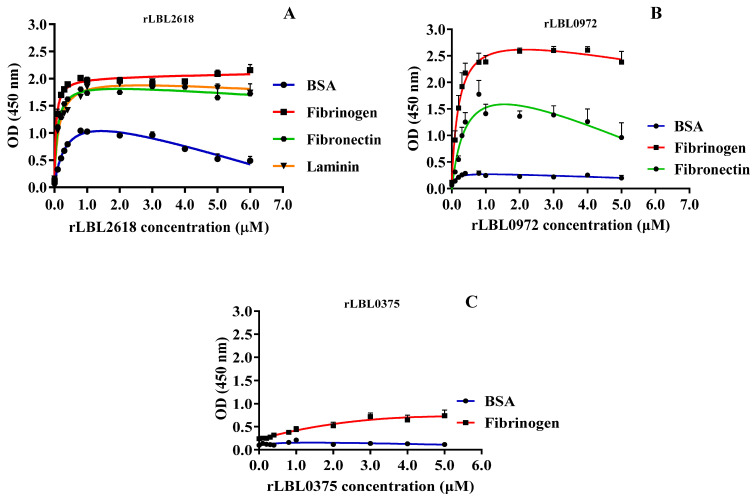
(**A**–**C**): Binding saturation curves of recombinant leptospiral OMPs to selected host ligand molecules. Data represent the mean absorbance at 450 nm ± SEM of three independent experiments. The estimated equilibrium dissociation constant (Kd) of each protein was calculated as the half-maximal binding from a non-linear regression analysis.

**Figure 6 microorganisms-12-00245-f006:**
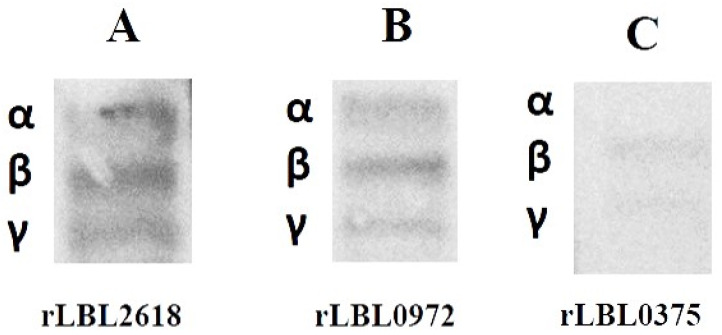
(**A**–**C**): Far-Western blot to detect binding of his-tagged OMPs to α-, β- and γ-chains of fibrinogen of OMPs. The molecular weights of fibrinogen chains α, β and γ are 64 kDa, 59 kDa and 48 kDa, respectively.

**Figure 7 microorganisms-12-00245-f007:**
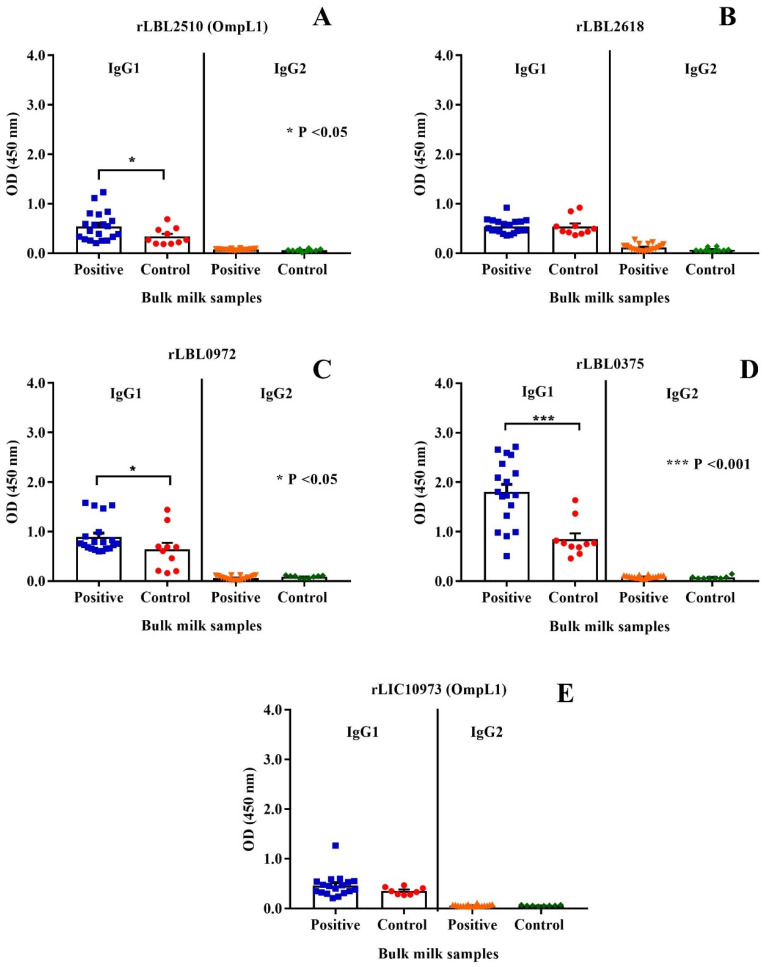
(**A–E**): Cattle milk antibody reactivity to leptospiral recombinant OMPs represented by scatter dot-plot graphs. Data represent the mean absorbance at 450 nm ± the standard error of the mean (SEM) of two independent experiments. The antibody titre against each OMP was compared between *Leptospira*-positive and -negative samples by Mann–Whitney U test (*** *p* < 0.001, * *p* < 0.05).

**Figure 8 microorganisms-12-00245-f008:**
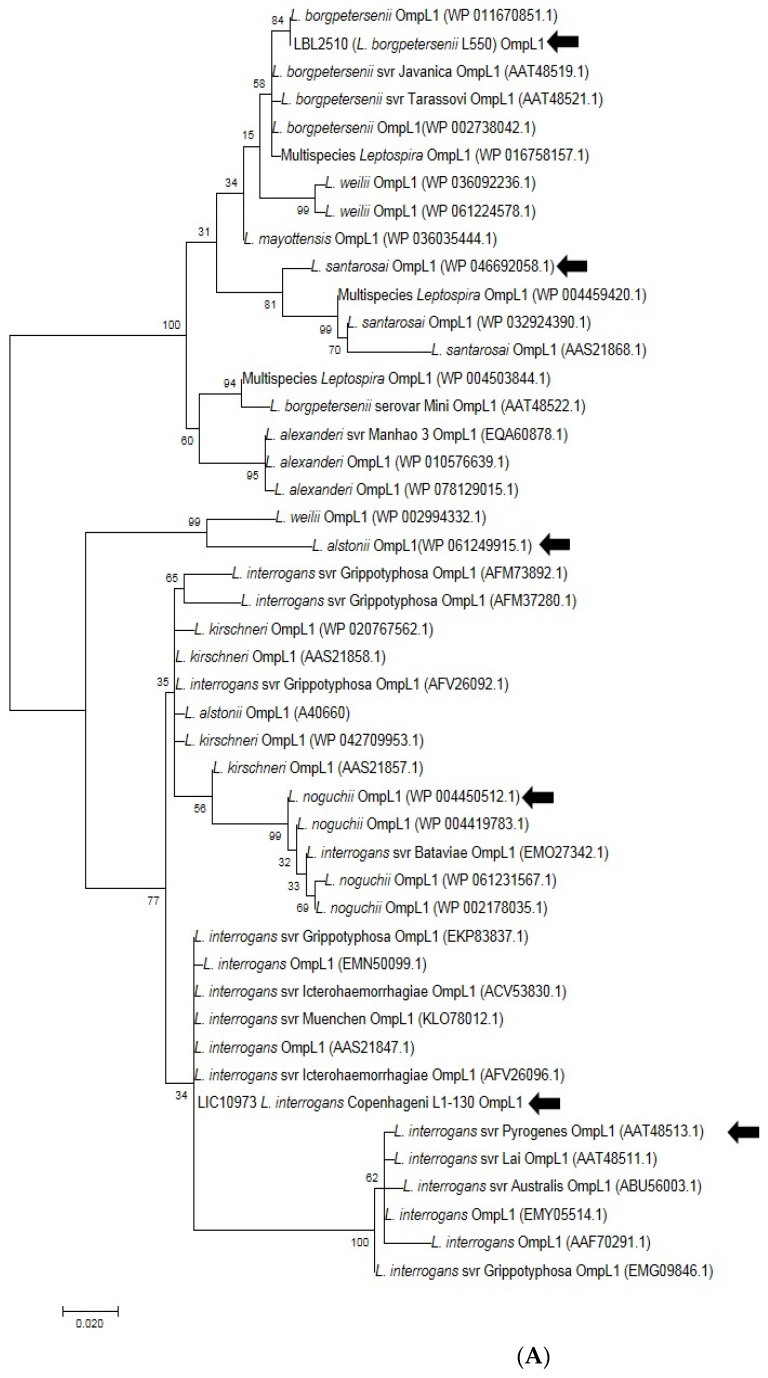
(**A**). Phylogenetic analysis of amino acid sequence of OmpL1 across various pathogenic *Leptospira* genomospecies and serovars by maximum likelihood method. Six OmpL1 genes, including *L. interrogans* serovar Hardjo-bovis L550 (LBL2510), *L. interrogans* serovar Pyrogenes (ATT48513.1), *L. santarosai* (WP046692058.1), *L. alstonii* (WP WP061249915.1) and *L. noguchii* (WP004450512.1P), from the deepest branches (indicated by black arrows) were selected for evaluation in this study. (**B**). Phylogenetic analysis of LBL2618 amino acid sequences across pathogenic *Leptospira* genomospecies by maximum likelihood method. Five LBL2618 genes, indicated by black arrows, including *L. borgpetersenii* serovar Hardjo-bovis L550 (LBL2618), *L. kirsheneri* (WP020778757.1), *L. santarosai* (WP004489573.1), *L. alstonii* (WP061250085.1) and *L. noguchii* (WP036066232.1), were selected for functional evaluation in this study.

**Figure 9 microorganisms-12-00245-f009:**
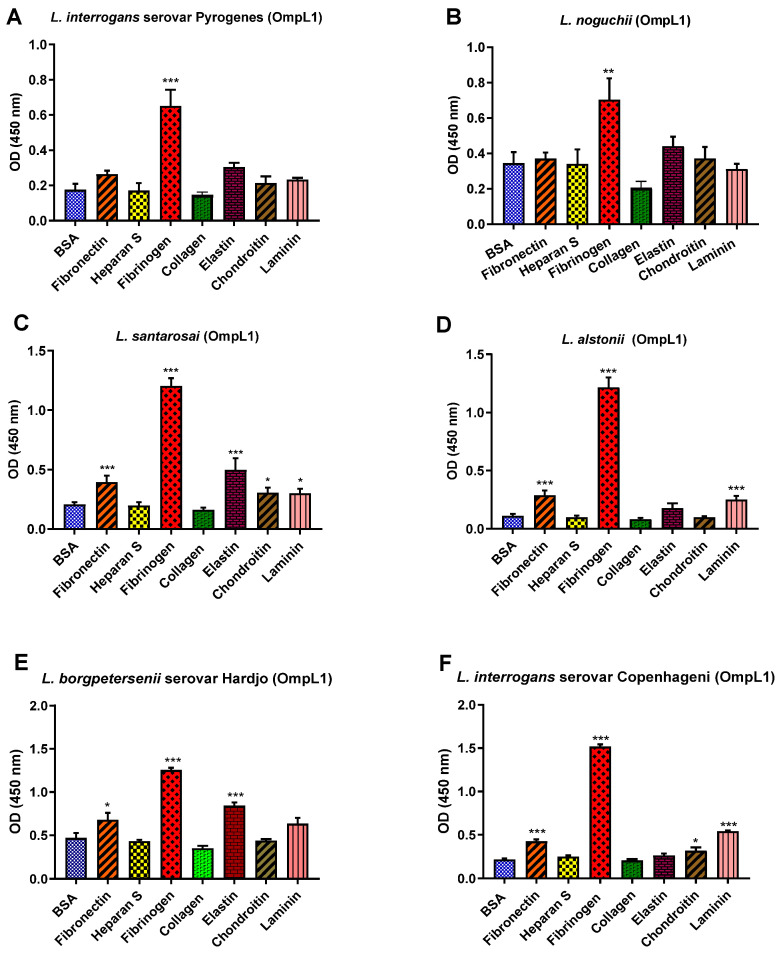
(**A**–**F**): Pathogenic leptospiral recombinant OmpL1 host component binding interactions. Data represent the mean absorbance at 450 nm ± the standard error and the mean (SEM) of three independent experiments. Statistical analysis was performed by one-way ANOVA and Dunnett’s Post Test using BSA as a control (*** *p* < 0.001, ** *p* < 0.01, * *p* < 0.05).

**Figure 10 microorganisms-12-00245-f010:**
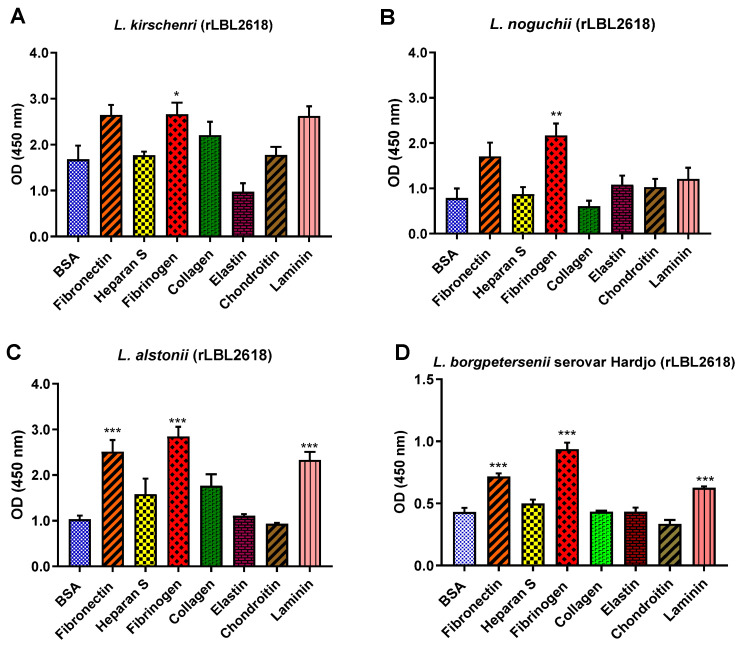
(**A**–**D**): Pathogenic leptospiral recombinant LBL2618 host component binding interactions. Data represent the mean absorbance at 450 nm ± the standard error and the mean (SEM) of three independent experiments. Statistical analysis was performed by one-way ANOVA and Dunnett’s Post Test using BSA as the control (*** *p* < 0.001, ** *p* < 0.01, * *p* < 0.05).

**Table 1 microorganisms-12-00245-t001:** Primer designs for putative OMPs subjected to expression studies.

Gene	Primer’s Sequence (Forward/Reverse) ^b^	Size (kb)
LBL1341	F: 5′ **cacc**CAACTTTGGACGCCGC 3′ R: 5′ TTAAAAACTTAAACCGCCCGA 3′	1.57
LBL0972	F: 5′ **cacc**AACGATGGAAACGAAAATTCTTC 3′ R: 5′ TTACGGGTTACAAGGCGC 3′	1.03
LBL1054	F: 5′ **cacc**GAACAAGTTGTAACCACGAAA 3′ R: 5′ TTAAAACTCTATTGTGGTTCTC 3′	1.39
LBL2618	F: 5′ **cacc**GAAAGGATCAGTATCGATGC 3′ R: 5′ TCAGAGATCATCACTGACG 3′	1.35
LBL2925	F: 5′ **cacc**GCTGAAAAAAAAGAGGAATCTGC R: 5′ TTATTGTTGTGGAGCGGAAG 3′	0.59
LBL0375	F: 5′ **cacc**CAAGAAGATTTGGATGAAAATCC 3′ R: 5′ TTATTTCTTGGCTGGAGGAG 3′	1.02
LBL2510 ^a^ (OmpL1)	F: 5′ **cacc**AAATCATACGCAATTGTAGGA 3′ R: 5′ TTAGAGTTCGTATTTATAGCCA 3′	0.89
LIC10973 ^a^ (OmpL1)	F: 5′ **cacc**AAAACATATGCAATTGTAGGATTTG 3′ R: 5′ TTAGAGTTCGTGTTTATAACCG 3′	0.89

^a^ OMP-positive controls used in this study include *L. interrogans* serovar Copenhageni L1-130 OmpL1 (LIC10973) and *L. borgpetersenii* serovar Hardjo-bovis OmpL1 (LBL2510) for host ligand binding and antibody titre experiments, respectively. ^b^ The **cacc** short overhang sequence was added at the beginning of the 5′ sequence to pair with the overhang sequence gtgg in the cloning entry vector (pENTR).

**Table 2 microorganisms-12-00245-t002:** Characteristics of *L. borgpetersenii* serovar Hardjo-bovis strain L550 putative OMP-encoding genes identified from in silico analysis of leptospiral genomes.

Locus Tag	Molecular Weight	Signal Peptide		β-Barrel Prediction				SPAAN Adhesin Prediction ^d^
	Size (kDa)	SPI/SPII	Cleavage Site	BOMP ^a^	PRED-TMBB ^b^ (Yes/No) with Scores		MCMBB ^c^	
LBL1341	61	SPI	IQA-QL	1	Yes	2.94	0.015	Yes
LBL0972	36	SPII	FAG-CA	0	Yes	2.88	0.024	Yes
LBL1054	50	SPI	THA-EQ	0	Yes	2.95	0.025	No
LBL2618	49	SPII	SQA-ER	0	No	3.02	0.008	Yes
LBL2925	20	SPII	SSA-EK	0	Yes	2.94	0.00	No
LBL0375	37	SPI	LVA-QE	0	Yes	2.92	0.016	Yes
LBL2510 ^e^ (OmpL1)	31	SP1	LSA-KS	0	Yes	2.91	0.024	Yes
LIC10973 ^e^ (OmpL1)	31	SPI	LSA-KT	1	Yes	2.90	0.024	Yes

^a^ BOMP: Proteins predicted to be integral OMPs were classified 0 to 5, where 0 did not find the protein to be an integral OMP, whilst 1 to 5 predicted integral OMPs, where 1 is the least reliable and 5 the most reliable prediction. ^b^ PRED-TMBB: Proteins predicted to be transmembrane OMPs based on the threshold level. Threshold below 3.00 indicates protein is likely β-barrel. ^c^ MCMBB: Based on a 1st order Markov Chain model, interpreting hydrophilic–hydrophobic residue order, where a score higher than 0 indicates a protein is β-barrel, whereas a score of 0 or lower indicates a protein is not β-barrel. ^d^ SPAAN: Proteins predicted to function as adhesins (based on the default cut-off probability score of 0.51). ^e^ OMP-positive controls used in this study included *L. interrogans* serovar Copenhageni L1-130 OmpL1 (LIC_10973) and *L. borgpetersenii* OmpL1 (LBL_2510).

**Table 3 microorganisms-12-00245-t003:** Binding interactions between the putative recombinant OMPs and host molecules.

Recombinant Protein	Host Ligand Molecules [Mean K_d_ (μM) ± SEM]						
	Fibronectin	Laminin	Heparin Sulphate	Chondroitin	Fibrinogen	Collagen	Elastin
rLBL2618	0.10 ± 0.02	0.14 ± 0.01	ND	ND	0.05 ± 0.01	ND	ND
rLBL0972	0.44 ± 0.21	ND	ND	ND	0.20 ± 0.06	ND	ND
rLBL0375	ND	ND	ND	ND	NS	ND	ND
rLIC10973 (LIC OmpL1)	0.81 ± 1.30	1.46 ± 5.44	ND	NS	0.92 ± 1.38	ND	ND

Apparent K_d_ for host molecule binding as the mean concentration of recombinant OMP at half-maximal binding. Abbreviations: ND: not determined, NS: not saturated.

**Table 4 microorganisms-12-00245-t004:** Correlation analysis of IgG1 antibody reactivities to each recombinant OMP.

Recombinant OMP/r Values	rLBL2510 (OmpL1) ^a^	rLBL2618	rLBL0972	rLBL0375	rLIC10973 (OmpL1) ^b^
rLBL2510 (OmpL1) ^a^		r = 0.01 NS	r = 0.01 NS	r = 0.02 NS	r = 0.41 (*p* < 0.0001)
rLBL2618			r = 0.80 (*p* < 0.0001)	r = 0.35 (*p* < 0.001)	r = 0.004 NS
rLBL0972				r = 0.50 (*p* < 0.0001)	r = 0.02 NS
rLBL0375					r = 0.04 NS
rLIC10973 (OmpL1) ^b^					

A summary matrix of r values of IgG1 titres of all recombinant OMPs against one another with corresponding *p* values. ^a^ OmpL1 of *L. borgpetersenii* serovar Hardjo-bovis L550; ^b^ OmpL1 of *L. interrogans* serovar Copenhageni L1-130; NS: not significant.

**Table 5 microorganisms-12-00245-t005:** Apparent Kd for host molecule binding as the mean concentration of recombinant variant OMP at half-maximal binding.

Leptospira Species	Binding to Host Molecules (K_d_) (Micromolar)							
	Annotation	Fibronectin	Laminin	Heparan Sulphate	Chondroitin	Fibrinogen	Collagen	Elastin _a_
*L. borgpetersenii* svr Hardjo L550	OmpL1	2.50 ± 1.77	ND	ND	ND	0.45 ± 0.11	ND	0.82 ± 0.35
*L. interrogans* svr Pyrogenes	OmpL1	ND	ND	ND	ND	0.86 ± 0.22	ND	ND
*L. noguchii*	OmpL1	ND	ND	ND	ND	0.78 ± 0.17	ND	ND
*L. santarosai*	OmpL1	1.37 ± 1.47	NS	ND	NS	0.30 ± 0.07	ND	1.72 ± 1.00
*L. alstonii*	OmpL1	0.13 ± 0.08	0.54 ± 0.37	ND	ND	0.21 ± 0.12	ND	ND
*L. interrogans* svr Copenhageni L1-130	OmpL1	1.27 ± 1.76	6.05 ± 18.5	ND	NS	1.54 ± 2.02	ND	ND
*L. borgpetersenii* svr Hardjo L550	rLBL2618	0.08 ± 0.01	0.15 ± 0.02	ND	ND	0.05 ± 0.01	ND	ND
*L. kirschneri*	rLBL2618	ND	ND	ND	ND	0.01 ± 0.00	ND	ND
*L. noguchii*	rLBL2618	ND	ND	ND	ND	0.04 ± 0.02	ND	ND
*L. alstonii*	rLBL2618	0.06 ± 0.03	0.05 ± 0.02	ND	ND	0.02 ± 0.01	ND	ND

^a^ Bovine elastin; ND, not determined as no significant binding identified from binding screen; NS, not saturable.

## Data Availability

Data are contained within the article and Appendix A.

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
