# Peer review of "Characterisation of Putative Outer Membrane Proteins from Leptospira borgpetersenii Serovar Hardjo-Bovis Identifies Novel Adhesins and Diversity in Adhesion across Genomospecies Orthologs"

_microorganisms, 2024, doi:10.3390/microorganisms12020245_

Round 1

Reviewer 1 Report

Comments and Suggestions for Authors

Leptospira spp. possess abundant outer membrane proteins (OMPs), which are believed to play a role as an adhesin and virulence factor. OMPs of Leptospira interrogans, a major pathogen of leptospirosis, have been well investigated so far, but this study focused on those of Leptospira borgpetersenii. Their in-silico search of putative OMP genes in the L. borgpetersenii genome and biochemical experiments showed the adhering ability of some recombinant L. borgpetersenii OMPs. The manuscript is well-written generally, though some problems are found in figures and data explanations. I hope my comments listed below will be useful to improve the manuscript.

Figure 2) The titles and values in the axes are too small. Please enlarge their font size. Figures S1 and S1 need the same modifications.

Figure 5) Why did some data (blue in A, and red and green in B) decrease with the protein concentration?

The sizes of fonts and symbols, and the shape of symbols are different among A, B, and C. Regarding the symbol shape, “Fibronectin” in A should be changed to an upward triangle, and “Fibrinogen” in C should be changed to a square as used in A and B.

Table 4 and lines 424-425) The authors stated, “Antibodies to all newly identified putative OMPs correlated with each other as represented by moderate to very strong r values (ranging from +0.35 - +0.80, P <0.001, P 425 <0.0001).” The results r = 0.5 and r = 0.35 are not “very strong”, which are moderate and weak, respectively. Please modify the explanation and discussion about the results.

Figure 8B) The black arrows are misaligned.

Figures 9 and 10) The colors for the representation of the host components are different among figures. To remove misunderstanding, the same color should be used for the same components. In my opinion, discrimination by color is not necessary, because the host components are described in the horizontal axis clearly.

Reviewer 2 Report

Comments and Suggestions for Authors

The manuscript "Characterisation of putative outer membrane proteins from Leptospira borgpetersenii serovar Hardjo-bovis identifies novel adhesins and diversity in adhesion across genomospecies orthologs" presents a comprehensive and well-structured study. The study's innovative approach to identifying and characterizing OMPs in Leptospira is commendable and presents significant potential for understanding leptospiral pathogenesis and host-pathogen interactions. However, I would like to offer the authors some specific comments and suggestions to enhance the clarity, depth, and scientific rigor of the paper:

1.      Please, expound on the criteria and thresholds used in the bioinformatics algorithms for predicting putative OMP genes. Clarity here will enhance reproducibility.

2.      The exclusion criterion based on >50% sequence identity with L. biflexa proteins is interesting. Could you provide a rationale for this specific threshold?

3.      Please clarify why the BL21-AI strain was chosen for expression and how its characteristics affect protein yield and folding.

4.      More details on the interpretation of CD spectroscopy data would be beneficial, especially how it correlates with the expected secondary structure of OMPs.

5.      Elaborate on the rationale behind the selection of specific host ligand components for the study.

6.      Please provide more details on the ELISA protocol modifications and their implications on the assay sensitivity and specificity.

7.      Discuss how the binding saturation curves were interpreted and the significance of these findings in the context of leptospiral pathogenesis.

8.      Please describe how the far-western blotting method specifically confirms the interaction between OMPs and fibrinogen chains.

9.      Clarify the criteria for selecting the 30 cattle bulk milk samples and how they represent the broader cattle population.

10.  Please discuss the power of the statistical tests used, considering the sample size and variability of the data.

11.  Please justify the choice of the pET-21a (+) expression vector for the construction of OMP variants.

12.  Please elaborate on the choice of expression conditions, particularly the combination of L-arabinose and IPTG, and its impact on protein expression levels and solubility.

13.  Please clarify the criteria used for selecting these six uncharacterised genes. Providing the rationale behind this selection could enhance the reader's understanding of their potential importance.

14.  Please discuss the potential reasons for the variable expression yields of the novel genes and the positive controls.

15.  The CD spectroscopy findings are intriguing. Could you elaborate on how these structural features might relate to the function of these OMPs?

16.  The observation of different fold states in OmpL1 is interesting. Discuss how these states might affect the protein’s function or immunogenicity.

17.  Please elaborate on the potential biological significance of the observed binding specificities of the OMPs to various host components.

18.  Please discuss the potential implications of the varying degrees of binding affinity to fibrinogen components among the OMPs.

19.  Please provide insights into how the observed antibody reactivities correlate with potential protective immunity or pathogenesis in bovine leptospirosis.

20.  Please discuss how the observed sequence diversity among OmpL1 and LBL2618 variants might influence their functional roles in the pathogen.

21.  Please address the failure of one variant (L. santarosai rLBL2618) to express and its potential impact on the study's conclusions.

22.  Please reflect on how the differences in binding affinities to host molecules among OmpL1 and LBL2618 variants might affect their roles in host-pathogen interactions.

Finally, addressing these comments would strengthen the scientific rigor and clarity of the findings, enhancing the paper's contribution to the field of microbial pathogenesis and host-pathogen interactions.
